# High-resolution global shipping emission inventory by Shipping Emission Inventory Model (SEIM)

Wen Yi [1]†, Xiaotong Wang [2]†, Tingkun He [1], Huan Liu [1*], Zhenyu Luo [1], Zhaofeng Lv [1], Kebin He [1]

[1] State Key Joint Laboratory of ESPC, School of Environment, Tsinghua University, Beijing 100084, China

[2] Key Laboratory of Beijing on Regional Air Pollution Control, Beijing University of Technology, Beijing 100124, China

[*] Correspondence to: liu_env@tsinghua.edu.cn (H. Liu)

† These authors contributed equally to this work

**Abstract:** The high-resolution ship emission inventory serves as a crucial dataset for various disciplines including atmospheric science, marine science, environmental management, etc. Here, we present a global high spatiotemporal resolution ship emission inventory at a resolution of $0.1° \times 0.1°$ for the years 2013, 2016-2021, generated by the state-of-the-art Shipping Emission Inventory Model (SEIMv2.2). Initially, the annual 30 billion Automatic Identification System (AIS) data underwent extensive cleaning to ensure data validity and accuracy in temporal and spatial distribution. Subsequently, integrating real-time vessel positions and speeds from AIS data with static technical parameters, emission factors, and other computational parameters, SEIM simulated ship emissions on a ship-by-ship, signal-by-signal basis. Finally, the results were aggregated and analyzed. In 2021, the ship activity dataset established based on AIS data covered 109.3 thousand vessels globally (101.4 thousand vessels reported by the United Nations Conference on Trade and Development). Concerning the major air pollutants and greenhouse gases, global ships emitted 847.2 million tons of $CO_2$, 2.3 million tons of $SO_2$, 16.1 million tons of $NO_x$, 791.2 kilo tons of CO, 737.3 kilo tons of HC (Hydrocarbon), 415.5 kilo tons of primary $PM_{2.5}$, 61.6 kilo tons of BC (black carbon), 210.3 kilo tons of $CH_4$, 45.1 kilo tons of $N_2O$ in 2021, accounting for 3.2% of $SO_2$, 14.2% of $NO_x$, and 2.3% of $CO_2$ emissions from all global anthropogenic sources, based on the Community Emissions Data System (CEDS). Due to the implementation of fuel-switching policies, global ship emissions of $SO_2$ and primary $PM_{2.5}$ saw a significant reduction of 81.3% and 76.5% in 2021 compared to 2019, respectively. According to the inventory results, the composition of vessel types contributing to global ship emissions remained relatively stable through the years, with container ships

consistently contributing ~ 30% of global ship emissions. Regarding vessel age distribution, the emission
contribution of vessels built before 2000 (without Tier standard) has been declining, dropping to 10.2%
in 2021, suggesting that even a complete phase-out of these vessels would have limited potential for
reducing NOx emissions in the short term. On the other hand, the emission contribution of vessels built
after 2016 (meeting Tier III standard) kept increasing, reaching 13.3% in 2021. Temporally, global ship
emissions exhibited minimal daily fluctuations. Spatially, high-resolution emission characteristics of
different vessel types were delineated. Patterns of ship emission contributions by different types of
vessels vary among maritime regions, with container ships predominant in the North and South Pacific,
bulk carriers predominant in the South Atlantic, and oil tankers prevalent in the Arabian Sea. The
distribution characteristics of ship emissions and intensity also vary significantly across different
maritime regions. Our dataset, which is accessible at https://zenodo.org/records/11069531 (Wen et al.,
2024), provides daily breakdown by vessel type and age is available for broad research purposes, and it
will provide a solid data foundation for fine-scale scientific research and shipping emission mitigation.
**1    Introduction**

44        Ships carry over 80% of global trade volume(Development, 2023). Employing heavy fuel oil, ships

emit significantly more atmospheric pollutants than diesel cars each year(Corbett et al., 1999; Endresen
et al., 2003; Jasper Faber, 2020). Existing studies indicate that ship emissions of atmospheric pollutants
and greenhouse gases have important environmental and climatic impacts on multiple spheres of the
Earth (Browse et al., 2013; Chen et al., 2020; Diamond, 2023; Zhang et al., 2021). In terms of air quality,
ships are regarded as a major source of $PM_{2.5}$ pollution in coastal cities (Liu et al., 2024; Luo et al., 2023).
Recent studies show that although ship emissions have decreased due to stricter control measures in
recent years, shipping-related mortality associated with long-term $PM_{2.5}$ exposure in Chinese coastal
areas increased by 11.4% from 2016 to 2020 as populations migrate towards coastal cities (Luo et al.,
2024). Ship emission-induced sulfur oxides aerosols, significantly influence local climates (Liu et al.,
2016; Yuan et al., 2022). With the increasing navigation in the Arctic, ship emissions of black carbon
have become a focal point of research and policy debates regarding their impact on the polar ice surface
(Stephenson et al., 2018; Zhang et al., 2019). Regarding their impact on marine organisms, anthropogenic
emissions account for over 80% of the utilizable nitrogen deposition in the ocean, with maritime
emissions contributing to 15% of global $NO_x$ emissions (Zhang et al., 2021). Given these facts,
characterizing ship emissions is crucial for fundamental research in atmospheric, marine, and climatic
sciences, etc.
The characterization of ship emissions through emission inventories stands as a pivotal and effective
methodology within maritime emissions research (Liu et al., 2016; Liu et al., 2024; Wang et al., 2021).
Over the past 30 years, with the improvement of ship activity data collection mechanisms, the
establishment of ship emission inventories has gradually shifted from the "top-down" approach, based
on fuel or power consumption statistics and empirical parameters, to the "bottom-up" approach, based
on high spatiotemporal resolution shipping trajectory data (Eyring et al., 2010; Jasper Faber, 2020; Liu
et al., 2016). Currently, the establishment of high spatiotemporal resolution ship emission inventories
based on Automatic Identification System (AIS) data has become the most popular tool for scientific
research and policy management in the field of ship emissions (Johansson et al., 2017; Kramel et al.,
2021; Wang et al., 2021). AIS consists of onboard equipment, shore-based and satellite-based receivers.
During navigation, the onboard equipment transmits AIS signals every 2 seconds to several minutes,
which are received by terrestrial or satellite-based AIS receivers and then transmitted in-time to servers
for storage. AIS messages record the ship's unique identifier and high-frequency dynamic information
that changes continuously as the vessel progresses, including the vessel's MMSI code, IMO number,
signal transmission time, ship's position (longitude and latitude), over-ground speed, operational status,
draft, and destination, among others. Leveraging real-time ship speed derived from AIS data along with
vessel technical specifications such as deadweight tonnage and design speed, and emission factors, it is
feasible to model instantaneous ship emissions and then aggregate them at a defined spatiotemporal
granularity, thereby constructing a high-resolution emission inventory dataset. The advantage of this
method is that the derived emission inventory does not rely on external spatiotemporal allocation
parameters, but retains accurate spatiotemporal information of ship emissions from AIS data (Liu et al.,
2016). However, the challenge lies in the difficulty of processing AIS data, the complexity of simulating
instantaneous ship emissions, and the significant computational resources required (Chen and Yang,
2024). Currently, the mainstream international ship emission inventory models based on AIS data include
the Ship Traffic Emission Assessment Model (STEAM)(Jalkanen et al., 2012; Johansson et al., 2017) ,
the Shipping Emission Inventory Model (SEIM), International Maritime Organization (IMO) emission
inventory model (Jasper Faber, 2020), Maritime Transport Environmental Assessment Model
(MariTEAM) (Kramel et al., 2021), etc.
In this study, we established a $0.1° × 0.1°$ global daily ship emission inventory for the years 2013,
2016-2021 based on SEIMv2.2. This dataset covers five air pollutants ($NO_x$, $SO_2$, $PM_{2.5}$, CO, HC) and
four greenhouse gases ($CO_2$, $CH_4$, $N_2O$, BC). Due to rigorous quality control, the ship emission inventory
established by SEIM possesses high information density, allowing for analysis across multiple
dimensions, such as fleet structure and spatiotemporal characteristics. Initially, we conducted meticulous
data cleaning and rigorous quality control on the commercially obtained global ship AIS data to establish
a reliable ship activity dataset. Subsequently, employing the latest emission factor and real-time engine
power simulation methods for ships, SEIMv2.2 computed instantaneous ship emissions, integrating
multiple quality control techniques such as interpolation processing for sparse routes and safety margin
considerations to ensure the accuracy of ship emission simulation. Finally, we aggregated ship emissions
from different temporal and spatial scales, as well as from different types and ages of ships. The derived
high-resolution global shipping emission inventory could serve as input data for climate or atmospheric
chemistry models.
The next section will elucidate the methodology and factors employed in establishing our high-
resolution ship emission inventory. Section 3.1 compares our results with previous global ship emission
inventories. Section 3.2 analyzes the temporal sequence of global ship emissions. Section 3.3 examines
the spatial distribution characteristics of global ship emissions. Section 4 provides information regarding
our dataset and data availability. Finally, Section 5 presents the conclusion.
**2     Methods**
**2.1     Ship Emission Inventory Model (SEIM)**
**2.1.1     General principles**
The Shipping Emission Inventory Model (SEIM) was first established by (Liu et al., 2016) based on the
idea of disaggregated dynamic method. Driven by AIS data, combined with each vessel's registration
information, SEIM realized real-time, vessel-by-vessel shipping emission simulations. SEIM is suitable
for the establishment of multi-scale shipping emission inventories with applications on regions (Liu et
al., 2016; Wang et al., 2021) and ports (Fu et al., 2017). SEIM has undergone two major updates:
SEIMv2.0 (Wang et al., 2021) and SEIMv2.2 (this study). Compared to SEIMv2.0, SEIMv2.2 features
three key improvements. (1) IMO numbers are employed as the primary identifier to match AIS data and
Ship Technical Specifications Database (STSD), and for those that cannot be matched, MMSI (Maritime
Mobile Service Identity) codes are used as the secondary identifier. We found that the matching rate of
the ship archive database established in previous years could significantly decrease when applied to new
years. This is because when ships are leased or AIS equipment is replaced, the MMSI code often changes,
while the IMO code remains constant. Therefore, using the IMO code as the first-choice identifier ensures
more accurate matching of AIS data and static ship information. See Sect. 2.1.2 for details. (2) The
formula for calculating the main engine load has been revised to include parameterized correction
schemes for draft, meteorological conditions, and hull fouling. Additionally, a main engine load
maximum limit of 98% is set to consider the navigation safety of ships. Refer to Sect. 2.1.3 for further
details. (3) The ship emission factors are comprehensively updated according to the Fourth Green House
Gases Study by IMO (Jasper Faber, 2020), and a black carbon calculation module has been integrated.
This update also integrates the Emission Control Area (ECA) module correction module directly into the
calculation process, rather than applying it as a post-process adjustment. Detailed methods are provided
in Sect. 2.1.4. During the development of SEIMv2.2, SEIMv2.1 was derived, which only updated the
emission factors compared to SEIMv2.0. Generally, the technical scheme of SEIMv2.2 is illustrated in
Fig. 1.

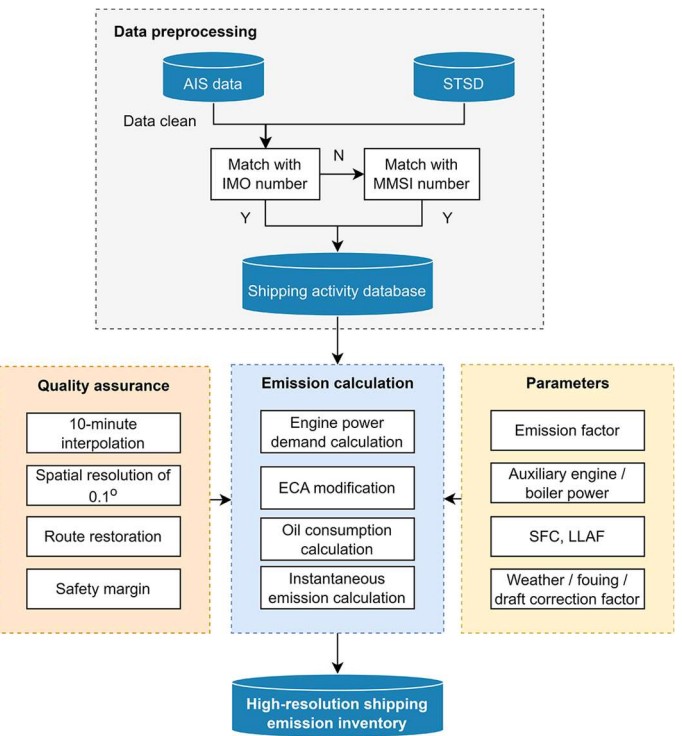


**Figure 1: The technical scheme for SEIMv2.2**

The calculation process and principles of SEIMv2.2 could be described as follows: Firstly, the original

AIS data collected are subjected to cleaning, missing data filling, etc., to establish a well-cleaned dynamic

AIS database. Secondly, IMO or MMSI codes are used as unique identifiers to match AIS data with the

STSD, which provides essential technical parameters such as vessel type, deadweight tonnage, main

engine power rating, design speed, etc., to establish a comprehensive shipping activity database. For the

details of STSD, refer to (Wang et al., 2021). This study has incorporated information on newly built

vessels from 2019 to 2021, obtained from the Lloyd's Register, into the established STSD. Thirdly, a

series of parameters such as emission factors, auxiliary engine and boiler output power, specific fuel

consumption, low load adjustment factors, and weather and fouling factors are input for ship emission

simulation. Then, the model will calculate GHGs (Greenhouse gases) and air pollutant emissions for

every ship by every two subsequent AIS signals. The emissions from the main engine, auxiliary engine,

and boiler are simulated using the corresponding formulas presented as Equation (1) - (3).

$$E_{ME,i,n,p} = P_{ME,i,n} \times EF_{ME,i,p} \times LLAF_{i,n,p} \times \Delta T_{i,n} \times 10^{-6} \qquad (1)$$


$$E_{AE,i,n,p} = P_{AE,i,n} \times EF_{AE,i,p} \times \Delta T_{i,n} \times 10^{-6} \qquad (2)$$


$$E_{B,i,n,p} = P_{B,i,n} \times EF_{B,i,p} \times \Delta T_{i,n} \times 10^{-6} \qquad (3)$$

Where the subscripts ME, AE, and B represent the main engine, auxiliary engine, and boiler, respectively;
$i$ represents individual ship; $n$ represents the $n$-th AIS signals in the sequence, and the total number of
AIS signals transmitted by the ship $i$ could be expressed using Ni; $p$ represents species of GHGs or air
pollutants. As for the capital letters, $E$ represents the emissions of GHGs or air pollutants (unit: ton); EF
is the emission factors (unit: g/kwh); $P$ is the output power (unit: kw); $\Delta T$ is the time interval of two
subsequent AIS signals (unit: h); LLAF is the low load adjust factor, which is applied only when the
main engine load factor is lower than 20%, consistent with our previous work. The total emissions are
calculated by summing up the emissions from all main engines, auxiliary engines, and boilers, as shown
in Equation (4).

$$E_{i,p} = \sum_{n=1}^{N_i-1} E_{i,n,p} = \sum_{n=1}^{N_i-1} (E_{ME,i,n,p} + E_{AE,i,n,p} + E_{B,i,n,p}) \qquad (4)$$

During the real-time calculation, linear interpolation is applied to latitude and longitude displacement as
well as time intervals where the AIS time interval is greater than ten minutes. AIS latitude and longitude
are rounded to one decimal place to ensure a spatial resolution of 0.1° × 0.1°. In the Chinese coastal
region, route restoration technology is applied to restore routes crossing land, referred to (Wang et al.,
2021). If, due to anomalies in speed or other factors, the main engine load exceeds 100%, it is capped at
98% for safety navigation considerations. Finally, the high-resolution emission inventory generated by
SEIM could be aggregated and analyzed from various angles such as emission structure, temporal
variations, spatial distribution, etc., depending on study demands.

### 2.1.2 AIS data cleaning

AIS data provide high-density vessel activity data, including time, speed, and latitude-longitude coordinates. This study collected global shore-based and satellite-based AIS data, with an average annual signal count of approximately 30 billion. Due to irregular or erroneous information entry at ports or on vessels, as well as interference from complex marine environments, weather conditions, and terrain, AIS data may suffer from errors, duplicates, and losses. To enhance the accuracy of emission inventory calculations, this study conducted meticulous cleaning of AIS data. Firstly, to ensure data validity, we filter AIS message records that met all of the following conditions: 1) Annual AIS signal count greater than 10; 2) Speed over ground less than 50 knots; 3) Longitudes ranged from -180° to 180° and latitudes ranged from -90° to 90°; 4) The timestamp of AIS signals within the target year.

Secondly, for temporal anomaly cleaning, signals with excessively long time intervals are filtered out. According to the International Convention for the Safety of Life at Sea (SOLAS), vessels are required to maintain continuous transmission of AIS signals throughout the year, except for specific reasons permitted by regulations. Therefore, vessels theoretically maintain a high frequency of signal transmission during navigation. Taking the year 2021 as an example, Fig. S 3 reveals a distinct bimodal distribution of original AIS signals, with peaks occurring around 300 seconds and 2 hours. The peak of around 300 seconds corresponds to the high-frequency interval for shore-based AIS equipment to receive signals, while the peak of around 2 hours mainly originates from satellite AIS signals. Statistical analysis shows that 99.6% of signal time intervals are within 8000 seconds. However, the total duration of these AIS signal intervals accounts for only 16% of all AIS signals, indicating the presence of extremely long consecutive signal intervals in the AIS data. This may be due to vessel docking for repairs or AIS equipment malfunctions. To minimize uncertainty in emission calculations caused by these signals, this study filters out signals with time intervals exceeding 7 days, which are not included in the emission calculations.

Thirdly, for spatial distribution anomalies cleaning, AIS data distributed on land are filtered out. Spatial distribution of the annual original AIS signal (Fig. S 4) revealed a significant number of signals deviating from shipping routes near the 0° and 120° meridians, located over Asia, Europe, and Africa. Further analysis indicated that such abnormal signal points are caused by misaligned field information or data loss. To minimize the interference of these abnormal signals on emission calculations, this study

employs the following cleaning methods: 1) For ships matched by IMO codes, signals with speeds >50
knots or consecutive latitude-longitude spans >20° are removed; 2) For ships matched by MMSI codes,
signals with speeds >40 knots or latitude-longitude spans >8° are excluded; 3) Signals located on land
areas are excluded. Fig. S 4 simultaneously presents the spatial distribution of the cleaned AIS signals,
revealing that signals on land near the 0° and 120° meridians have been eliminated while retaining signals
on major navigable rivers in North America, South America, and Eurasia.
Fourthly, to ensure the reliability of ship technical parameter data matching, the AIS data are
subsequently aggregated and identified with IMO numbers and MMSI numbers. AIS data comprises
static AIS data and dynamic AIS data. Static AIS data include time, MMSI numbers, IMO numbers, etc.,
but do not contain latitude and longitude information. Dynamic AIS data contain MMSI numbers and
latitude and longitude information. This study integrates the static AIS data and dynamic AIS data by
matching their common MMSI numbers. It is found that approximately 84% of the MMSI numbers in
dynamic information could be matched with a unique valid seven-digit IMO number. However, some
ships may change their MMSI codes multiple times within a year. Data with MMSI code changes more
than 10 times are excluded from emission calculations.
**2.1.3    Engine power demand**
Engine power demand is crucial for emission calculations. For the main engine, its real-time output
power is related to the main engine load, which can be depicted in real-time by changes in the ship's
speed-over-ground obtained from AIS data. According to the propeller law, the main engine load factor
is the cube of the ratio of the ship's actual speed to its design speed (Liu et al., 2016). Additionally, some
studies indicate that factors such as draft, hydrological and weather conditions, and hull fouling also
influence the main engine load (Chen and Yang, 2024; Emmens et al., 2021; Fu et al., 2022). Regarding
draft factors, the (Jasper Faber, 2020) corrects the main engine load using real-time draft data from AIS.
However, draft fields in commercial AIS data are often manually recorded by crew members, leading to
low accuracy and a large number of zero values. As for hydrological and weather conditions, wind and
waves could increase engine power demand through friction and shear resistance. (Johansson et al., 2017)
adopts a method based on a real-time ship heading and weather field, which requires substantial
computational resources and introduces greater uncertainty by the weather field. Additionally, the
accumulation of micro- and macro-organisms on ship surfaces increases power demand to overcome
resistance, and existing studies often use fixed parameters to correct the influence. This study introduces
parameterization schemes to correct the influence of draft, weather, and hull fouling. Based on the ships'
payload utilization calculation algorithm in (IMO, 2015), this study estimates the average drafts for
different types of vessels, with specific values provided in Table S1. The correction coefficients for
weather influences ($\eta_w$) are based on (Jasper Faber, 2020), also presented in Table S1. The correction
coefficient for fouling influences ($\eta_f$) is set to 0.917. Specifically, the formula for calculating the real-
time power of the main engine in SEIMv2.2 can be found in equation (5).

$$P_{ME,i,n} = P_{ref,i} \times LF_{i,n} = \frac{P_{ref,i} \times \left(\frac{D_i}{D_{ref,i}}\right)^{0.66} \times \left(\frac{v_{i,n}}{v_{ref,i}}\right)^3}{\eta_w \times \eta_f} \tag{5}$$

Where $P_{ref,i}$ represents the maximum engine output power (unit: kw) of the main engine of the
ship i; $LF_{i,n}$ represents the main engine load factor of the ship i at the n-th AIS signals in the sequence.
$D_i$ represents the average draft; $D_{ref,i}$ represents the designed draft; $v_{i,n}$ represents the speed-over-
ground (unit: knot) of the ship i at the n-th AIS signals in the sequence; $v_{ref,i}$ represents the design
speed (unit: knot) of ship i, obtained from the static technical profiles; $\eta_{w,i}$ represents the weather
correction factor and $\eta_{f,i}$ represents the fouling correction factor, both of which are unitless.
For auxiliary engines and boilers' power demand, this study adopts the recommended values from the
IMO Fourth and Third Greenhouse Gas Study reports. Due to the lack of information, this study did not
consider the impact of other auxiliary devices on board, such as solar panels, wind sails, waste heat
recovery systems, and carbon capture, utilization and storage (CCUS) systems, on vessel energy
consumption. These systems are not significant contributors to overall vessel energy consumption
currently (Dnv, 2022). However, with the ongoing trends of energy efficiency improvements, the impact
of these systems on vessel energy utilization could be transformative in the future (Kersey et al., 2022).
**2.1.4 Emission factors**
The emission factors applied by SEIMv2.0 is mainly based on the Third IMO GHG Study (Smith,
2014) as well as the National Standard for General Diesel Fuel of the People's Republic of China (Wang
et al., 2021). In this study, we updated the emission factors based on the Forth IMO GHG Study (Jasper
Faber, 2020), and therefore accompanying technical modification. Firstly, emission factors of
conventional air pollutants ($SO_2$, $NO_x$, $PM_{2.5}$, CO, HC) and GHGs ($CO_2$, $CH_4$, $N_2O$) were updated. $CO_2$,
$SO_2$, and $PM_{2.5}$ are considered typical species whose emission factors are highly dependent on the
chemical component of fuels. Table. S2 represents the emissions factors based on fuel consumption for
$CO_2$, $SO_2$, and $PM_{2.5}$. Energy-based emission factors are calculated based on fuel-based emission factors
as well as specific fuel consumptions (SFC, unit: kwh/kg fuel), using Equation (6):

$$EF_e = EF_f \cdot SFC \qquad (6)$$

SFC represents the fuel consumption per unit of work performed by a ship, mainly decided by the
fuel calorific value (kwh/kg fuel) and engine efficiency (%). During the operation of ships, energy
efficiency could be considered as a quadratic function of the load factor of the main engine, generally
with the optimal load factor of 80%. Equation (7) is applied to calculate the SFC for main engines based
on the SFC under the optimal operating condition ($SFC_{base}$) and main engine load of the ship $i$.

$$SFC_{ME,i} = SFC_{base,ME,i} \cdot (0.455 \cdot LF_i^2 - 0.71 \cdot LF_i + 1.28) \qquad (7)$$

Generally, newer ships have a lower $SFC_{base}$ than older ships due to the improvement of engine
and auxiliary engine efficiency (Sou et al., 2022). The LNG fleet also has a lower $SFC_{base}$ value than
conventional fuel. SFC of auxiliary engines and boilers ($SFC_{AE|B,i}$) is not subject to the main engine load,
so $SFC_{AE|B,base,i}$ is directly applied with no main engine load adjustment. Values of $SFC_{base}$ are
exhibited in Table. S 3.
Combining Table. S 2 and Equations (1), and (2), energy-based emission factors for the main
engines of $CO_2$ and $SO_2$ as a function of the main engine load could be derived, as exhibited in Fig. S 1.
It could be noted that although Marine Gas Oil (MGO) has a higher carbon content compared to Heavy
Fuel Oil (HFO), its lower SFC results in a lower energy-based $CO_2$ emission factor. Fig. S 2 illustrates a
comparison between two algorithms employed in SEIMv2.0, which utilizes uniform emission factors for
all operational conditions, and SEIMv2.2, which incorporates load-dependent emission factors. We
selected a typical oil tanker with dead-water tonnage of 7562 tons and examined its hourly carbon
emissions from July 1 to July 15, 2019. It is evident that, at anchorage or berth, the hourly emissions
estimated by SEIMv2.2 are generally higher than or equal to those of the previous SEIMv2.0. When the
vessel is cruising, however, the overall emissions calculated by SEIMv2.2 are relatively lower compared
to SEIMv2.0. Emission factors of other air pollutants and GHGs in this study are shown in Table S 4 and
Table S 5.
**2.2    Data source and quality control**
**2.2.1    AIS data coverage**
The AIS-observed data obtained in this study amounted to approximately 30 billion/year, while the
processed AIS signals after cleaning and interpolation averaged about 4-5 billion per year, with an
average annual operating time of approximately 5-7 million hours, as shown in Table 1. In comparison
to (Johansson et al., 2017), the AIS signal volume in this research is slightly lower, possibly due to
comprehensive quality control measures in data reduction and filtering, which removed a significant
number of signals with inadequate validity, abnormal time or spatial distribution, and insufficient
reliability. It can be observed that with the increasing prevalence of AIS equipment, the quantity of AIS
signals is on the rise. However, the operating time does not necessarily increase in proportion to the
signal quantity. The operating time decreased by 3.8% in 2020 compared to 2019 and increased by 4.5%
in 2021 compared to 2020, probably influenced by the pandemic,

**Table 1: Annual AIS signals and operating time after data cleaning**

| Year | 2013 | 2016 | 2017 | 2018 | 2019 | 2020 | 2021 |
|---|---|---|---|---|---|---|---|
| AIS signals, billion | 1.6 | 4.1 | 4.6 | 5.0 | 5.0 | 5.0 | 5.5 |
| Operating time, million hour | 225.8 | 578.5 | 643.6 | 700.9 | 693.8 | 667.3 | 697.6 |

Taking 2021 as an example, the spatial distribution of AIS signals after cleaning and time interpolation is illustrated in Fig. 2. The spatial coverage of cleaned AIS signals is extensive, with signals primarily concentrated along major shipping routes such as the coastal regions of East Asia, the Malacca Strait-Cape of Good Hope route, the Mediterranean, and the Black Sea routes, effectively depicting the trajectories of major shipping lanes.

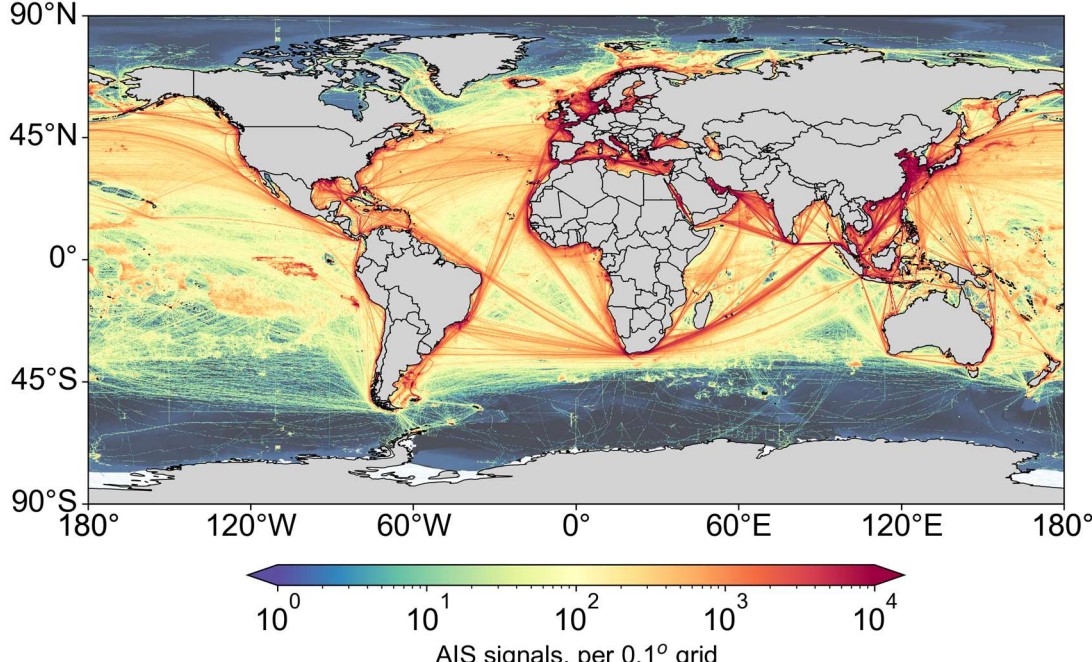

**Figure 2: The spatial distribution of global AIS signals in 2021. Maps are made with Natural Earth.**

### 2.2.2    Global fleet composition

Table 2 presents the global fleet structure obtained through matching AIS data with STSD in 2021. This study covers 14 vessel types, including major cargo vessels, passenger ships, and fishing vessels, along

with a category labeled "others," comprising research vessels, rescue ships, and work vessels, among
others. Since the "others" category primarily consists of small coastal vessels, its contribution to
emissions is minor. It's important to note that United Nations Conference on Trade and Development
(UNCTAD) definition of "others" differs from this study's categorization. According to UNCTAD 2021
classification, other ships include liquefied petroleum gas carriers, liquefied natural gas carriers, parcel
(chemical) tankers, specialized tankers, reefers, offshore supply vessels, tugboats, dredgers, cruise ships,
ferries, and other non-cargo ships. Moreover, it should be noted that the fleet obtained in this study
comprises vessels with certain activity levels (annual AIS signals exceeding 10), whereas UNCTAD
statistics do not consider vessel activity. This discrepancy might lead to comparatively lower results in
this study.
Overall, the discrepancies between the global fleet statistics in this study and those of UNCTAD are not
substantial. In terms of vessel numbers, this study reached 109.3 thousand in 2021, slightly higher than
UNCTAD's 101.4 thousand. The total deadweight tonnage amounts to 1989.9 million tons, slightly lower
than UNCTAD's 2136.2 million tons. Among the fleet obtained in this study, vessels matched by IMO
numbers reach 62.9 thousand, contributing 57.5% of vessel count and 95.4% of deadweight tonnage.
Vessels matched by MMSI numbers constitute a larger proportion in count (42.5%), yet make a smaller
contribution in deadweight tonnage (4.6%), predominantly consisting of fishing vessels (which
contribute 87.9% to the vessel count matched by MMSI numbers). There's a noticeable difference in the
quantity of general cargo ships and oil tankers. However, in terms of total tonnage, the container ships,
general cargo ships, bulk carriers, and oil tankers show no significant differences (below 10%) with
UNCTAD, ensuring the reliability of global ship emission calculations and emission structure analysis.

**Table 2: Comparison of the global fleet structure of this study and the UNCTAD statistics in 2021. The fleet**
**analyzed in this study was filtered to include vessels with an annual AIS signal count greater than 10.**

| Vessel type | Number of vessels, thousand | | | | Total deadweight tonnage, million ton | | | |
|---|---|---|---|---|---|---|---|---|
| | This study | | | UNCTAD | This study | | | UNCTAD |
| | match with IMO number | match with MMSI number | total | | match with IMO number | match with MMSI number | total | |
| Auto Carrier | 0.8 | 0.1 | 0.8 | | 15.3 | 0.9 | 16.2 | |
| Bulk Carrier | 11.1 | 0.6 | 11.7 | 12.3 | 842.3 | 27.6 | 869.9 | 913.2 |
| Chemical Tanker | 4.7 | 0.3 | 5.0 | | 100.5 | 4.5 | 105.0 | |
| Container | 4.2 | 0.3 | 4.5 | 5.4 | 218.8 | 14.4 | 233.2 | 281.8 |
| Cruise | 0.2 | 0.1 | 0.2 | | 0.1 | 0.0 | 0.1 | |
| Fishing ship | 5.1 | 40.8 | 45.8 | | 4.0 | 5.8 | 9.7 | |
| General Cargo | 6.9 | 0.8 | 7.7 | 20.0 | 66.4 | 4.4 | 70.7 | 77.9 |
| LNG | 0.3 | 0.0 | 0.3 | | 19.3 | 0.2 | 19.5 | |
| LPG | 1.2 | 0.1 | 1.2 | | 19.0 | 0.8 | 19.9 | |
| Miscellaneous | 12.3 | 1.4 | 13.7 | | 76.1 | 6.5 | 82.6 | |
| Ocean Tug | 7.0 | 0.9 | 7.9 | | 13.4 | 1.6 | 15.0 | |
| Oil Tanker | 5.4 | 0.4 | 5.8 | 11.5 | 505.3 | 22.2 | 527.5 | 619.3 |
| Reefer | 0.5 | 0.0 | 0.5 | | 3.5 | 0.2 | 3.7 | |
| Ro Ro | 3.0 | 0.6 | 3.6 | | 11.8 | 1.7 | 13.5 | |
| Others | 0.5 | 0.0 | 0.5 | 52.2 | 3.1 | 0.0 | 3.1 | 243.9 |
| Total | 62.9 | 46.4 | 109.3 | 101.4 | 1899.1 | 90.8 | 1989.9 | 2136.2 |


## 3 Results

### 3.1 Total global shipping emissions

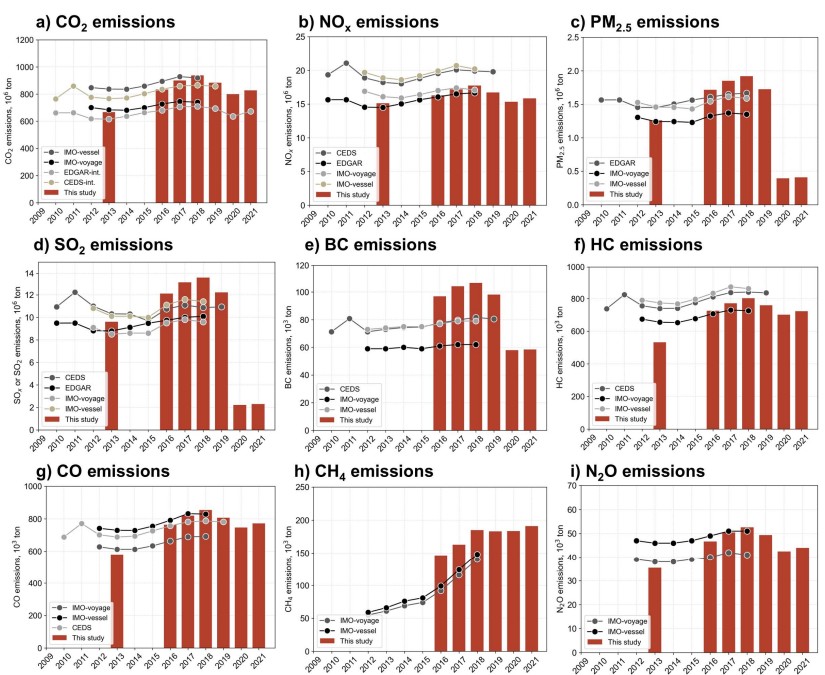

**Figure 3: Global trends in shipping emissions from 2010 to 2021.** Data source: IMO (Jasper Faber, 2020), where IMO-Voyage results were calculated based on a voyage-based method and IMO-Vessel on a vessel-based algorithm. Community Emissions Data System (Mcduffie et al., 2020); Emissions Database for Global Atmospheric Research (Crippa, 2021).

We established a multi-year global ship emissions inventory with temporal resolution of day and spatial resolution of 0.1° using the SEIM model for the years 2013 and 2016-2021. Figure 3 summarizes this study and open-source dataset of major atmospheric pollutants and greenhouse substances emitted by global shipping over the past decade. The ship emission calculation method employed in this research, which is AIS-based, aligns with those utilized in the EDGAR (Emissions Database for Global Atmospheric Research) inventory and the Fourth IMO GHG Study released in 2020, while the CEDS inventory is established based on a top-down fuel-based approach (Mcduffie et al., 2020). Methodologically, our study is more comparable to the research conducted by EDGAR and IMO, with the results from the CEDS inventory as a reference. It is important to note that the SEIM model has undergone two major version updates. The data of different versions are presented in Figure S3. Figure 3 presents the integrated results for ease of comparison with other studies. Specifically, the results for 2013 are based on SEIMv1.0, while those for 2016-2020 are based on SEIMv2.1 and those for 2021 are

based on SEIMv2.2. Due to slight differences between the two versions (Fig. S5), both of which include
data for 2020, the total emissions for 2020 in SEIMv2.2 were adjusted to match those in SEIMv2.1. The
growth rate for 2021 was kept consistent to ensure that the data for both versions align in 2020. In terms
of annual emission totals, this study's results show similarities in emission trends and total emissions
compared to well-known inventories such as EDGAR and IMO. For most species, this study's results
show higher annual growth rates compared to IMO and EDGAR studies. For instance, this study
estimates a 6.1% annual increase rate in global ship $CO_2$ emissions from 2016 to 2018, while IMO's
study indicates only a 1.4% annual increase for its "vessel-based" results and 0.9% for its "voyage-based"
results. In 2019 and 2020, influenced by international trade conflicts and the global pandemic, this study
estimates a 5.8% and 9.5% year-on-year decrease in global ship $CO_2$ emissions for 2019 and 2020,
respectively. In contrast, the year-on-year decrease rate estimated by EDGAR inventory is 2.1% and 8.4%
for 2019 and 2020, respectively. Differences between studies may stem from factors such as AIS data
quality, coverage of static information, and factors considered in emission calculations. In 2020, the
global implementation of the fuel-switching policy led to a significant reduction in the sulfur content of
ship fuel. According to SEIM, in 2020 relative to 2019, $SO_2$, $PM_{2.5}$, and BC emissions decreased by
81.9%, 77.2%, and 40.9%, respectively. In 2021, following the recovery in global trade demand after the
pandemic, this study estimates an increase of 3.5% in global ship $CO_2$ emissions compared with 2020,
while EDGAR inventory estimates an increase of 5.9%. However, the latest data on ship's atmospheric
pollutants from other inventories only extend to 2019, which is insufficient for comparing emission
results with this study.

## 3.2    Temporal evolutions

### 3.2.1    Daily shipping emissions

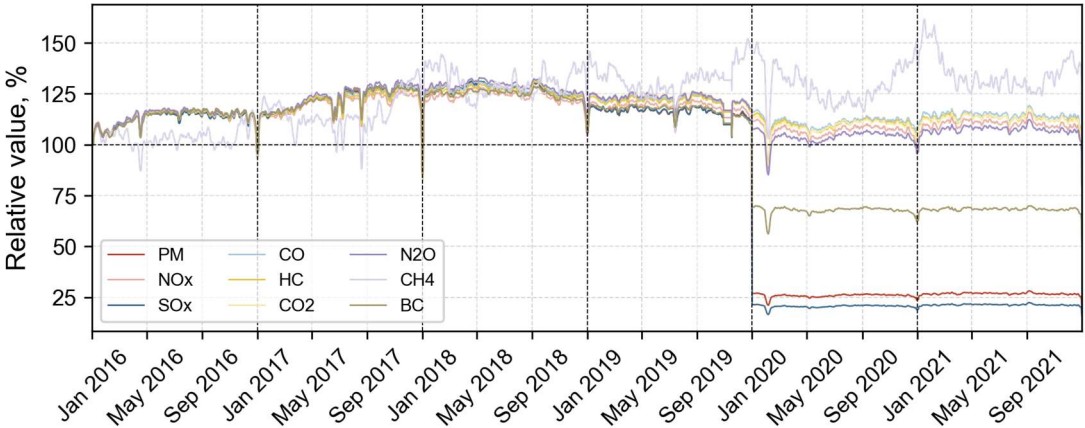

**Figure 4: Daily global shipping air pollutants and GHGs emissions from 2016 to 2021.** With January 1, 2016, as the reference point, the five-day moving average of daily relative emissions is displayed.

To compare the magnitude of changes in emissions of various atmospheric pollutants and greenhouse gases emitted by ships, Fig. 4 converts the daily ship emissions of 9 species into daily relative quantities taking the emission levels on January 1, 2016, as the reference point. Figure 4 reveals that, aside from occasional sharp declines or increases on certain dates, the daily variations in ship emissions are generally stable. This suggests that the emission simulations by the SEIM model exhibit continuity and stability. Ships typically cruise at constant speeds on high seas without significant diurnal or other periodic variations, so global ship emissions do not exhibit pronounced daily or seasonal fluctuations. Any anomalies such as sudden drops or spikes may be attributed to signal transmission anomalies in equipment or meteorological factors. In 2019, the reduction in ship $SO_2$ emissions compared to 2018 was slightly larger than that of other pollutants, probably attributed to the implementation of the domestic emission control area policy within 12 nautical miles of the Chinese coast, one of the world's busiest areas for shipping activities (Chen et al., 2017), which has also been demonstrated by Fig. 7c. From Fig. 4, finer temporal patterns can be observed, such as the gradual increase in emissions during the second half of 2017 and the subsequent decrease in ship emissions in 2019 as trade conflicts intensified. In 2020, the impact of the pandemic led to two phases of decline and recovery in global ship emissions. The 2020 global fuel-switching policy also led to a significant reduction in ship $SO_2$, $PM_{2.5}$, and BC emissions. Despite the implementation of NECA policy from 2016 to 2021 (IMO, 2023), the decline in ship $NO_x$

emissions is very slow, as shown in Fig. 4, which is due to the fact that the current fleet is still
predominantly composed of ships built before 2016 (accounting for more than 85%, as shown in Figure
6). The slow pace of fleet renewal makes it more challenging to achieve substantial reductions in $NO_x$
emissions from ships currently. It is worth noting that $CH_4$ emissions exhibit relatively large daily
changes and have been increasing throughout the six years. The primary source of $CH_4$ emissions is LNG
ships. The daily fluctuations in ship $CH_4$ emissions are mainly due to variations in LNG ship activities.
Although LNG ships are currently relatively few, their quantity is increasing as the demand for low-
carbon ships grows steadily (Gronholm et al., 2021).
**3.2.2    Multi-dimensional structure**

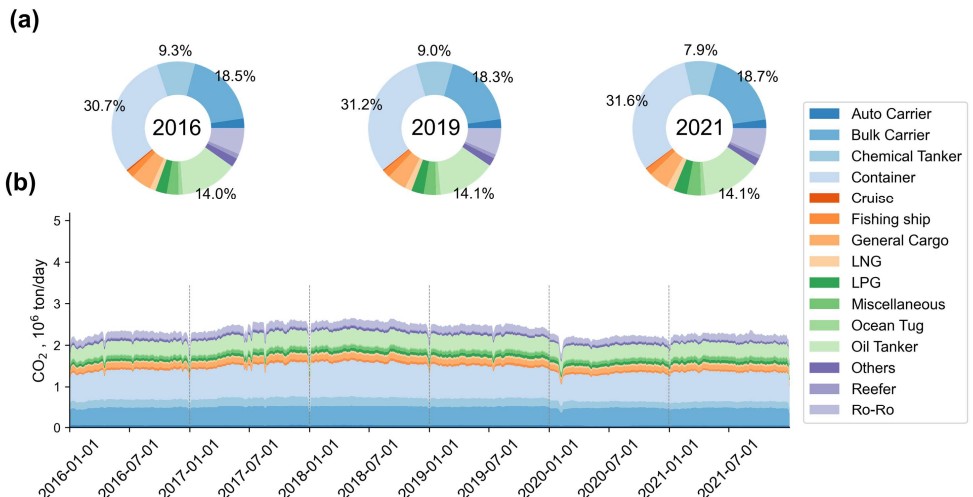

**Figure 5: Composition of global ship $CO_2$ emissions by vessel type from 2016 to 2021.** a) the percentage
contribution of emissions by vessel type for the years 2016, 2019, and 2021; b) the five-day moving average of
daily emissions for different vessel types.
Figure 5 displays the daily $CO_2$ emissions classified by vessel type. From 2016 to 2021, container
ships, bulk carriers, and oil tankers consistently contributed the most, accounting for 31.6%, 18.7%, and
14.1% of global ship $CO_2$ emissions in 2021, respectively. The contribution of container ships increased
from 30.7% to 31.6% from 2016 to 2021. Overall, there were no significant changes in the composition
of vessel types over the six years. Vessel types reflect the types of commodities transported by sea,
indicating the relative stability of the global maritime cargo structure.

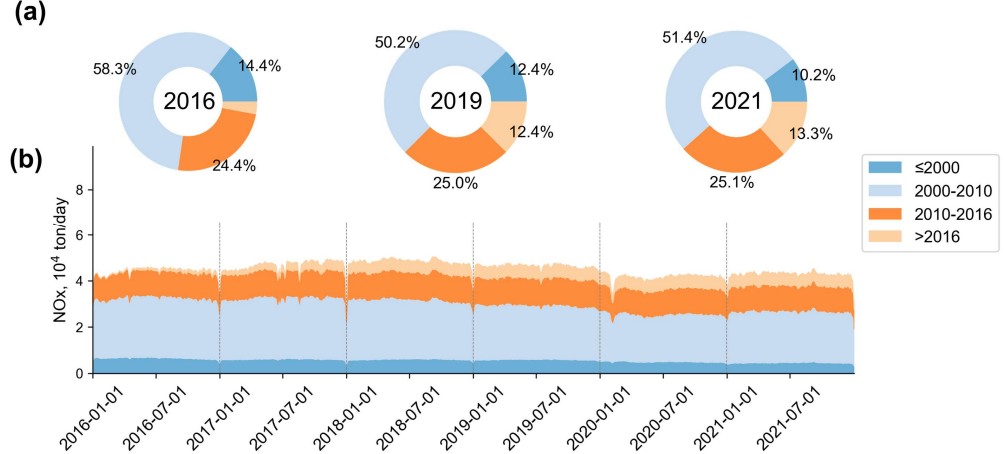

Figure 6: Composition of global ship $NO_x$ emissions by vessel construction year from 2016 to 2021. a) the percentage contribution of emissions by vessels constructed in different periods for the years 2016, 2019, and 2021; b) the five-day moving average of daily emissions for vessels constructed in different periods.

Figure 6 illustrates the daily $NO_x$ emissions composed by the vessel construction period. The construction year of vessels determines the $NO_x$ emission standards followed by their engine (IMO, 2008). From 2016 to 2021, vessels complying with Tier I standards (built during the year 2000-2010) consistently contributed over 50% of ship $NO_x$ emissions, while those complying with Tier II standards (built during the year 2010-2015) contributed approximately half of Tier I emissions. As the majority of ship $NO_x$ emissions come from Tier I- and Tier II-standard vessels, global ship $NO_x$ emissions are expected to remain at current levels in the short term without further control measures. However, it is noteworthy that the contribution of vessels over 20 years old (built-year $\leqslant$ 2000) to global ship $NO_x$ emissions has gradually decreased, from 14.4% in 2016 to 10.2% in 2021. Meanwhile, the contribution of newly built vessels (built-year $\geqslant$ 2016) to global ship $NO_x$ emissions has steadily increased from 3.5% in 2019 to 13.3% in 2021.

### 3.3    Spatial characteristics

### 3.3.1    High-resolution patterns

Based on latitude and longitude coordinates in AIS signals, the ship emissions dataset was spatially aggregated into grids, resulting in the global spatial distribution of ship emissions. Figure 7 depicts the $SO_2$ emissions from global ships in $0.1°\times0.1°$ grids. The regions with high intensity of ship $SO_2$ emissions include East Asia, South Asia, Europe, the Persian Gulf, the Mediterranean, and the western coast of Europe. The intensive ship emissions along major global shipping routes are clearly visible, such

as the routes connecting East Asia through the Malacca Strait, the Suez Canal, and the Strait of Hormuz
to Western European countries (the "Europe-Middle East-Far East route"), the Strait of Gibraltar, the
Strait of Hormuz, and the critical passage connecting the Pacific and Atlantic Oceans, the Panama Canal,
among others. Comparing the spatial distribution of ship $SO_2$ emissions in different years, noticeable
reductions in emissions are observed in ECAs such as North America, the Gulf of Mexico, the North Sea,
and the Baltic Sea comparing the ship $SO_2$ emissions distribution in 2013 and 2016. A significant
reduction in emissions is also observed in the Domestic Emission Control Area (DECA) comparing the
ship $SO_2$ emissions distribution in 2016 and 2019. In 2021, the implementation of the global low-sulfur
fuel policy resulted in a significant overall reduction in ship $SO_2$ emissions spatially compared with 2019.
The spatial distribution of ship $SO_2$ emissions in different years demonstrates that the SEIM v2.2
effectively responds to $SO_2$ emission control policies.

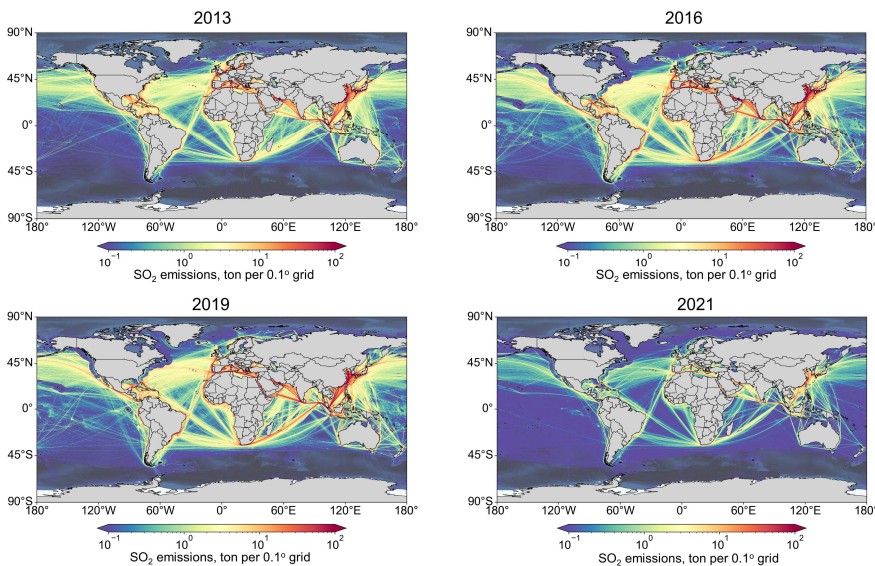


**Figure 7: Spatial distribution of global ship $SO_2$ emissions in different years**
**3.3.2    Spatial disparities of vessel composition**
Distinct disparities in spatial distribution are evident between freight vessels and non-freight
vessels in Fig. 8. Emissions from container ships, bulk carriers, and oil tankers are concentrated in major
international shipping lanes. In contrast, emissions from non-transport vessels such as fishing vessels are
more widely distributed in non-lane open sea areas. According to this study, in 2021, fishing vessels
contributed 1.6% to global ship $CO_2$ emissions, with their emissions mainly concentrated in the North
Sea, Baltic Sea, Yellow Sea, and South Pacific. In recent years, studies have utilized fine satellite data to
reveal significant fishing vessel activities that had not been publicly tracked worldwide (Paolo et al.,
2024). The emissions from those fishing vessels remain unknown. Therefore, the emissions from fishing
vessels presented in this study should be considered highly uncertain and are not discussed in the
following sections.

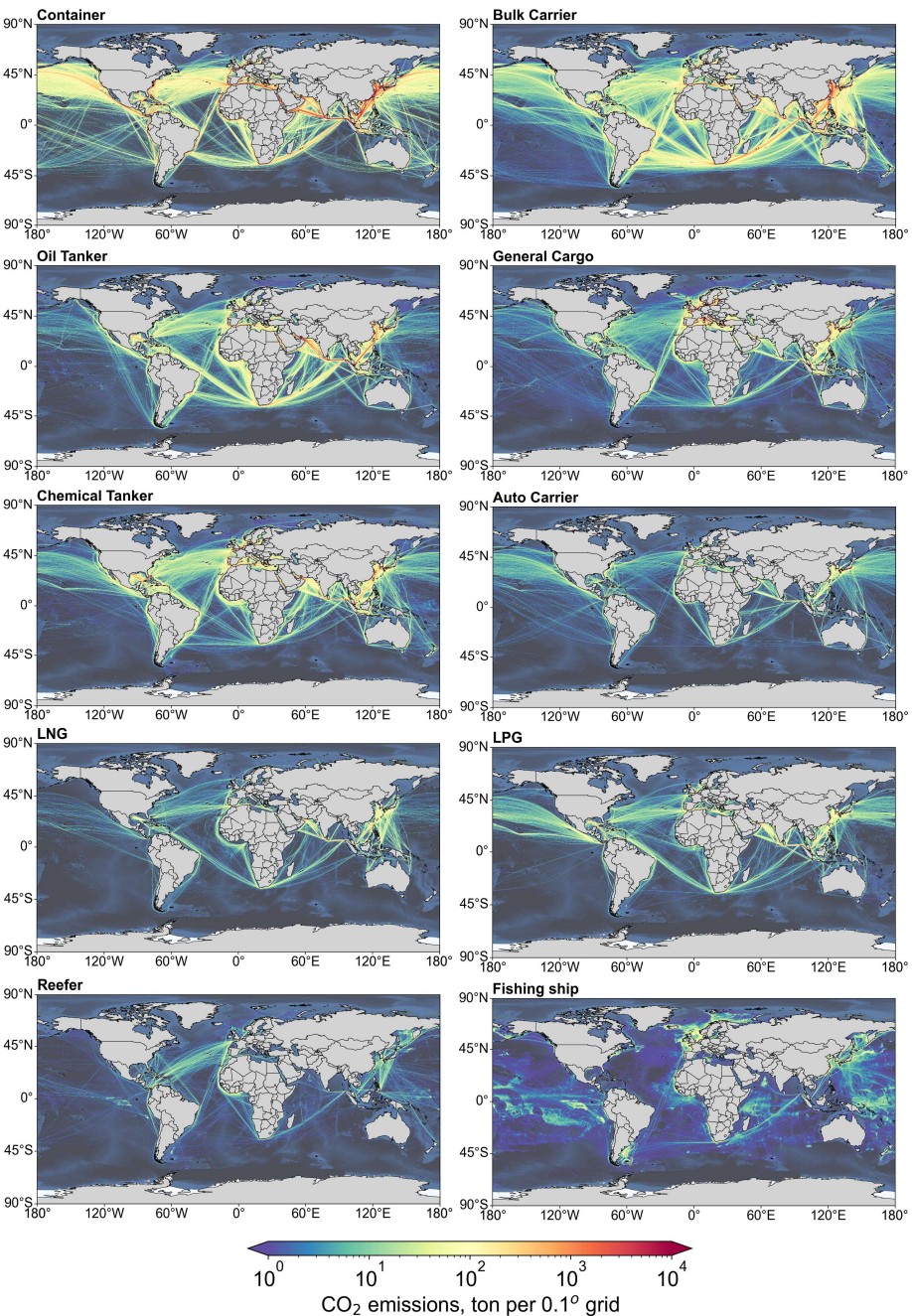


**Figure 8: Spatial distribution of CO$_2$ emissions from different types of vessels in 2021**
The spatial distribution of emissions varies across different vessel types, leading to disparities in the
composition of vessel types in different maritime regions. The division of global maritime regions is
based on the International Hydrographic Organization (IHO) standards (https://iho.int/). Figure 9
illustrates the composition of vessel types in the top 14 regions with the highest $CO_2$ emissions globally
in 2021, with their combined emissions accounting for almost 80% of the total global emissions. In the
pie charts for each region, vessel types contributing over 10% of emissions are labeled. It is observed
that container ships contribute significant emissions in the North Pacific Ocean, South China Sea, and
East China Sea, accounting for 49.4%, 37.4%, and 38.7% respectively, well above the global average
(31.6%). Regions with high contributions from bulk carriers are mainly distributed in the southern
hemisphere, such as the Indian Ocean (40.7%) and South Atlantic (34.9%). Ro-Ro vessels exhibit high
emissions proportions near Europe, with percentages of 38.7% in the Baltic Sea, 19.3% in the North Sea,
and 18.1% in the Mediterranean. Oil tankers contribute 28.5% of emissions in the Arabian Sea, probably
attributed to countries like Saudi Arabia and Iran, rich in oil and gas resources, generating substantial
shipping emissions during exports to other countries.

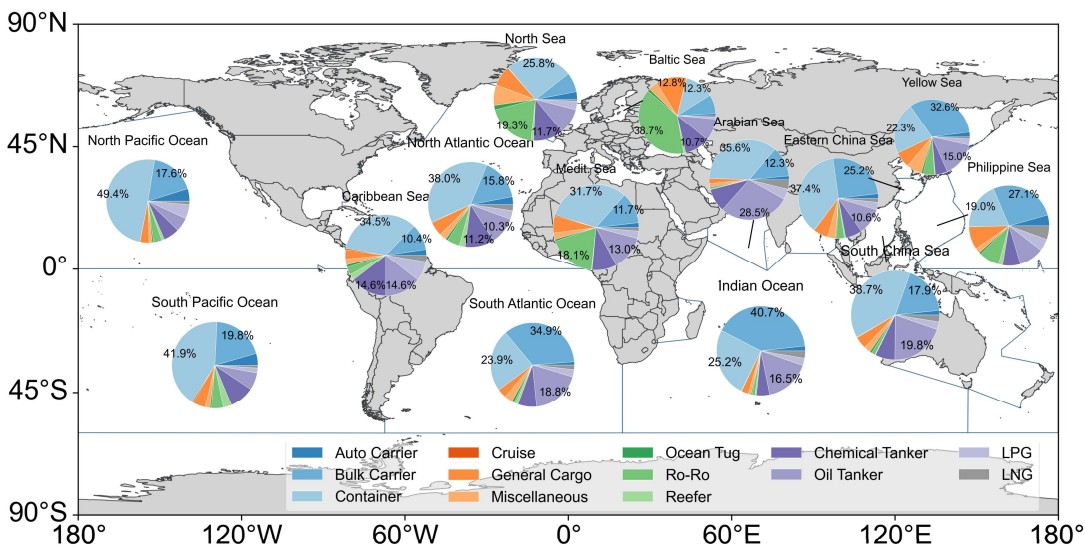


**Figure 9: Ship CO₂ emissions composition in different maritime regions globally in 2021 (excluding fishing**
**ships and others)**

### 3.3.3 Emission intensity

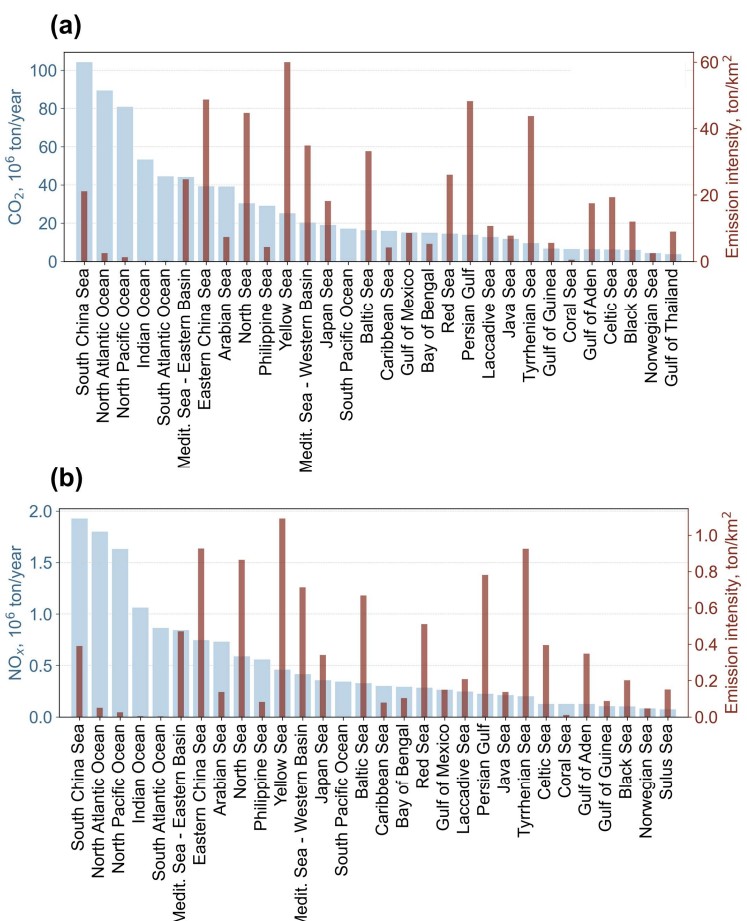

**Figure 10: Global ship (a) CO₂ and (b) NOₓ emission and emission intensities in different maritime regions in 2021**

Since the significant differences in the area of $0.1° \times 0.1°$ grids at different latitudes, ship emissions within each grid are standardized into emission intensity, i.e., emissions per unit area (unit: $ton/km^2$). Taking ship $CO_2$ emissions as an example, Fig. 10 illustrates the total ship $CO_2$ emissions and emission intensity in major maritime regions in 2021. The top 30 maritime regions with the highest $CO_2$ and $NO_x$ emissions, accounting for approximately 96% of the total global ship emissions, are listed and arranged in descending order of total emissions. It is important to note that the South/North Pacific, South/North Atlantic, and Indian Ocean cover a vast area (about 75% of the total maritime area), most of which have little or no ship navigation. Calculating the total average emission intensity for these regions would weaken their significance, so they are not discussed here. Among other maritime regions, the South China Sea has the highest total ship $CO_2$ and $NO_x$ emissions. As a vital route for maritime trade between East Asia Europe and Africa, the South China Sea exhibits prominent ship traffic density globally.

Additionally, the Eastern Mediterranean Basin, the Arabian Sea, the East China Sea, the Philippine Sea,
and the North Sea also have relatively significant total emissions. Generally, maritime regions with high
$CO_2$ emissions also have relatively high $NO_x$ emissions. Although the order of maritime regions with
lower emissions differs slightly, overall consistency is observed. There are significant differences
between maritime regions with high emissions and those with high emission intensity. The top five
maritime regions with the highest ship $CO_2$ emission intensity are the Yellow Sea, the Persian Gulf, the
East China Sea, the North Sea, and the Tyrrhenian Sea. The top five for $NO_x$ emission intensity are also
the same maritime regions. These regions are coastal areas or busy maritime routes with intensive ship
emissions, which warrants attention in environmental management in the future.
**4    Data Availability**
Shipping emission data described in this manuscript can be accessed at Zenodo under
https://zenodo.org/records/11069531 (Wen et al., 2024).
**5    Conclusions**
Utilizing the SEIM model, we developed a high-resolution ship emission inventory covering the
period from 2013 to 2016-2021 globally, encompassing 5 atmospheric pollutants ($NO_x$, $SO_2$, $PM_{2.5}$, CO,
HC) and 4 greenhouse gases ($CO_2$, $CH_4$, $N_2O$, BC). With a temporal resolution of day and spatial
resolution of $0.1^{\circ} \times 0.1^{\circ}$, our inventory revealed novel insights into global ship emission characteristics.
In terms of annual emissions, our inventory exhibits consistency in temporal trends and emission
magnitudes compared to mainstream inventory datasets including EDGAR, CEDS, and IMO. According
to this study and the global anthropogenic emission inventory by (Hoesly Rachel, 2024), ship emissions
contributed 12.3% of $SO_2$, 14.0% of $NO_x$, and 2.5% of $CO_2$ to global anthropogenic emissions in 2019,
and 3.2% of $SO_2$, 14.2% of $NO_x$, and 2.3% of $CO_2$ to global anthropogenic emissions in 2021. Over the
years, ship $CO_2$, $NO_x$, CO, HC, and $N_2O$ emissions showed a declining trend due to the impacts of the
2019 trade conflict (year-on-year decrease rate 5.4%-6.2%) and the 2020 pandemic (year-on-year
decrease rate 7.4%-13.8%), with a subsequent rebound in 2021 as international trade increased (year-on-
year increase rate 3.1%-3.6%). SOx, PM, and BC emissions were significantly influenced by gradually

implemented ECA policies and the 2020 low-sulfur fuel-switching policy. SOx and PM emissions in 2021 were 80.9% and 76.0% of those in 2019, and BC emissions were 38.7% of those in 2019. $CH_4$ emissions exhibited an increasing trend over the years, growing by 43.5% in 2021 compared to 2016.

Regarding emission composition, container ships consistently constituted the primary source of global ship $CO_2$ emissions, contributing over 30% annually and steadily increasing, followed by bulk carriers, oil tankers, and chemical tankers. The proportion of emissions contributed by new ships increased annually from 3.5% in 2016 to 13.3% in 2021. However, Tier I and Tier II ships still dominate ship $NO_x$ emissions. Currently, Tier III standards only apply to vessels operating in North American Emission Control Areas. Achieving a significant reduction in global ship $NO_x$ emissions still requires extensive advancements in ship engine technology and follow-up regulatory measures worldwide.

As for spatial characteristics, ship emissions were particularly significant in East Asia, South Asia, and Europe, with busy shipping routes such as the Western Europe-Middle East-Far East route, the Strait of Malacca, the Strait of Gibraltar, the Strait of Hormuz, and the Panama Canal showing the highest emission intensities. The regions with the highest $CO_2$ and $NO_x$ ship emission intensities were the Yellow Sea, the Persian Gulf, the East China Sea, the North Sea, and the Tyrrhenian Sea. These are not only areas with the highest emission intensity but also coastal regions with dense populations and ecosystems vulnerable to pollution. This suggests that these regions should be prioritized in environmental management efforts for improving air quality, protecting marine ecosystems, and climate mitigation. Furthermore, influenced by the types of commodities transported and the countries involved in trade, significant differences in ship emission characteristics exist across different regions. SEIM enables the analysis of the heterogeneity of spatial distributions of ship emissions. In terms of vessel type composition, container ships significantly exceeded the global average in ship $CO_2$ emissions contributions in the North Pacific, East China Sea, and South China Sea. Regions with high proportions of emissions from bulk carriers were mainly located in the Southern Hemisphere, such as the Indian Ocean, South Pacific, and South Atlantic. Emissions from oil tankers were high in the Arabian Sea and the Persian Gulf. The findings on the spatial heterogeneity of global ship emissions offer insights into region-specific management. In addition, since many high-emission regions include transboundary areas, such as the South China Sea and the Mediterranean, where maritime traffic connects multiple countries., effective mitigation in these regions will require international cooperation.

Although the complex quality control processes employed in this study, uncertainties still persist in
the aspects of AIS data accuracy, emission factors and so on. In the next steps, more work should be
done to reduce the uncertainties in bottom-up ship emission evaluation model, including integrating latest
methods and multi-source data to improve the accuracy of AIS data quality control, gathering more
studies on recent ship emission factors to cover more ship size and operating status, as well as involving
multiple data sources such as satellite data to validate the results.Overall, SEIM offers a globally multi-
year, high spatiotemporal resolution ship emission inventory that provides reliable and detailed data,
which could support foundational research across disciplines such as atmospheric science, environmental
science, and geoscience. Meanwhile, this dataset could also provide scientific support for facilitating
shipping emission mitigation in the future.
**Acknowledgments**
This work was supported by the National Natural Science Foundation of China (grant nos. 42325505),
National Key Research and Development Program of China (No. 2023YFC3705604 and No.
2022YFC3704200), and China Postdoctoral Science Foundation (No. 2023M730142).
**Author contributions**
W.Y. and X.W designed the research and wrote the manuscript, H.L. reviewed and revised the
manuscript, and T.H. contributed to the modelling. Z.L. and Z.L. contributed to data analysis. K.H. and
H.L. provided insights into the research design. All authors contributed to the writing. We also thank
Harvard-China Project on Energy, Economy and Environment and Professor Chris P. Nielsen from
Harvard University, for his suggested revisions to this manuscript.
**Competing interests**
The authors declare no competing interests.

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
