# Peer review of "High-resolution global shipping emission inventory by Shipping Emission Inventory Model (SEIM)"

_Earth System Science Data, 2024_

## Author Comment (AC1)

**Response to Reviewers #1's Comments**

**General Comment:**

I really enjoy reading this well-written and highly interesting manuscript presented by Yi et al. The authors use a Shipping Emission Inventory Model to carefully analyze and generate a powerful set of global emission inventory. The analytical process was thoroughly stepped through in the manuscript, and the data was well-presented. These figures and data would be useful to policy-makers and technology-developers, in addition to scientists in atmospheric and ocean sciences. I'd recommend publication with minor revision.

**Response:**

Thank you very much for your recognition. We improved our manuscript according to your suggestions and tried our best to address all the concerns in this revision.

**Comment #1:**

One small suggestion I have for the authors is that, the definitions of some terms could be better clarified so that the general audience who are not so familiar with the field can understand more easily. For example, I am not 100% sure what AIS signal means – does this mean how many ships there are in this dataset? If so, the authors can simply say this in the captions of Table 1 and Figure 2, and also define this term in the text.

**Response:**

AIS signals are generated by the Automatic Identification System (AIS) installed on ships. The system consists of onboard equipment, shore-based and satellite-based receivers. During navigation, the onboard equipment transmits AIS signals every 2 seconds to several minutes, which are received by terrestrial or satellite-based AIS receivers and then transmitted in-time to servers for storage. AIS messages record the ship's unique identifier and high-frequency dynamic information that changes continuously as the vessel progresses. The fields within an AIS message include the vessel's MMSI code, IMO number, signal transmission time, ship's position (longitude and latitude), over-ground speed, operational status, draft, and destination, among others. As such, the volume of AIS signals reaches billions, far exceeding the number of ships in the dataset. To better clarify the meaning of AIS signals, we have included an explanation of "AIS signals" in the manuscript.

**Revisions in manuscript:**

*Line 70-76*: AIS consists of onboard equipment, shore-based and satellite-based receivers. During navigation, the onboard equipment transmits AIS signals every 2 seconds to several minutes, which are received by terrestrial or satellite-based AIS receivers and then transmitted in-time to servers for storage. AIS messages record the ship's unique identifier and high-frequency dynamic information that changes continuously as the vessel progresses, including the vessel's MMSI code, IMO number, signal transmission time, ship's position (longitude and latitude), over-ground speed, operational status, draft, and destination, among others.

**Comment #2:**

Also, please define HC, BC, and other abbreviations that appear in the manuscript.

**Response:**

Regarding abbreviations, we conducted a review and reintroduced explanations for the following abbreviations: HC (Hydrocarbon), BC (Black Carbon), MMSI (Maritime Mobile Service Identity), EDGAR (Emissions Database for Global Atmospheric Research), and GHGs (Greenhouse Gases).

**Revisions in manuscript:**

*Line 21-26*: Concerning the major air pollutants and greenhouse gases, global ships emitted 847.2 million tons of $CO_2$, 2.3 million tons of $SO_2$, 16.1 million tons of $NO_x$, 791.2 kilo tons of CO, 737.3 kilo tons of HC (Hydrocarbon), 415.5 kilo tons of primary $PM_{2.5}$, 61.6 kilo tons of BC (black carbon), 210.3 kilo tons of $CH_4$, 45.1 kilo tons of $N_2O$ in 2021, accounting for 3.2% of $SO_2$, 14.2% of $NO_x$, and 2.3% of $CO_2$ emissions from all global anthropogenic sources, based on the Community Emissions Data System (CEDS).

*Line 116-118*: IMO numbers are employed as the primary identifier to match AIS data and Ship Technical Specifications Database (STSD), and for those that cannot be matched, MMSI (Maritime Mobile Service Identity) codes are used as the secondary identifier.

*Line 144-145*: Then, the model will calculate GHGs (Greenhouse gases) and air pollutant emissions for every ship by every two subsequent AIS signals.

*Line 335-340*: Figure 3 summarizes this study and open-source dataset of major atmospheric pollutants and greenhouse substances emitted by global shipping over the past decade. The ship emission calculation method employed in this research, which is AIS-based, aligns with those utilized in the EDGAR (Emissions Database for Global Atmospheric Research) inventory and the Fourth IMO GHG Study released in 2020, while the CEDS inventory is established

based on a top-down fuel-based approach (Mcduffie et al., 2020).

**Comment #3:**

*In addition, figures could have higher resolutions. Right now, it is hard to read the legends. Figure 3i, 'N2O' does not have the subscript.*

**Response:**

The figures in the manuscript have all been updated, now with a resolution of 300 dpi, according to the requirement of ESSD.

Particularly, Fig. 3 has been updated, with the previously non-subscripted "$N_2O$" in the title corrected and legends enlarged. The updated Fig. 3 has been provided in the ***Revisions in Manuscript*** below (Figure Q1). Other figures, which were only updated with enhanced resolution, without any content changes, has not been provided below.

**Revisions in manuscript:**

[Figure]

Figure Q1: Global trends in shipping emissions from 2010 to 2021. Data source: IMO (Jasper Faber, 2020), where IMO-Voyage results were calculated based on a voyage-based method and IMO-Vessel on a vessel-based algorithm. Community Emissions Data System (Mcduffie et al., 2020); Emissions Database for Global Atmospheric Research (Crippa, 2021).

---

## Author Comment (AC2)

**Response to Reviewers #2's Comments**

**General Comment:**

This manuscript is a study that established the global ship emissions inventories from 2013, 2016 to 2021 using the SEIMv2.2 model, providing a comprehensive analysis of the patterns of spatiotemporal variations in ship emissions throughout the years. The authors conducted a thorough cleaning of the global AIS data, correcting for spatial drift, excessively long-time intervals, and data misalignment, and employed compression techniques to make the data computable. This extraordinary work is quite challenging, but it ultimately ensures the quality of the data, allowing for a broad analysis of ship emissions from multiple angles, including the composition of ship types, age distribution, temporal changes, spatial variations, and analyses across different intersecting dimensions. The article's visualization of the data results is also very clear and intuitive. The dataset it provides could be useful for future scholars in the fields of atmosphere, ocean, and environment. Overall, this is a good paper that deserves to be published in ESSD. Nevertheless, some minor issues must be clarified.

**Response:**

Thank you very much for your recognition. We improved our manuscript according to your suggestions and tried our best to address all the concerns in this revision.

**Comment #1:**

First, in the analysis of temporal changes, the authors could include some discussion on international policies, particularly how recent fuel-switching regulations impact changes in international shipping emissions, especially within Emission Control Areas (ECA).

**Response:**

During 2016 to 2019, the international SECA, which includes the Baltic Sea, North Sea, North America, and Caribbean Sea, maintained a constant limit on sulfur content in marine fuels at 0.1% m/m (IMO, 2023). In parallel, China's domestic ECA policy (DECA) was gradually implemented and tightened, covering the water area 12 nautical miles away from the Chinese mainland's territorial sea baseline by 2019 (Wang et al., 2021). As shown in Figure 4, there was a slight decrease in $SO_2$ and $PM_{2.5}$ in 2019, reflecting the impact of Chinese DECA policy. Figure 7c further corroborates this observation. In 2020, the

significant decrease in $SO_2$, $PM_{2.5}$, and BC was attributed to the global fuel switch policy.

As for NECA, starting from January 1, 2016, ships constructed on or after that date and operating in the North America and Caribbean Sea emission control areas are required to comply with $NO_x$ Tier III standards set forth in regulation 13.5 of MARPOL Annex VI. Similarly, the Baltic Sea and North Sea require compliance with NOx Tier III standards for ships constructed on or after January 1, 2021, operating within these areas (IMO, 2023). In Fig. 4, the decline in $NO_x$ is very slow, which is due to the fact that the current fleet is still predominantly composed of ships built before 2016 (accounting for more than 85%, as shown in Figure 6). The slow pace of fleet renewal makes it more challenging to achieve substantial reductions in NOx emissions from ships currently.

In the revised manuscript, we have strengthened the connection between various sections (Fig. 4, Fig. 6 and Fig. 7) and enhanced the discussion on temporal changes related to ECA policies.

**Revisions in manuscript:**

*Line 379-382*: In 2019, the reduction in ship $SO_2$ emissions compared to 2018 was slightly larger than that of other pollutants, probably attributed to the implementation of the domestic emission control area policy within 12 nautical miles of the Chinese coast, one of the world's busiest areas for shipping activities (Chen et al., 2017), which has also been demonstrated by Fig. 7c.

*Line 386-391*: The 2020 global fuel-switching policy also led to a significant reduction in ship $SO_2$, $PM_{2.5}$, and BC emissions. Despite the implementation of NECA policy from 2016 to 2021 (IMO, 2023), the decline in ship $NO_x$ emissions is very slow, as shown in Fig. 4, which is due to the fact that the current fleet is still predominantly composed of ships built before 2016 (accounting for more than 85%, as shown in Figure 6). The slow pace of fleet renewal makes it more challenging to achieve substantial reductions in $NO_x$ emissions from ships currently.

**Comment #2:**

Second, could the authors add a paragraph discussing on the uncertainties and limitations of the model in the conclusion section? This could include a discussion on the accuracy of AIS data, the uncertainties in emission factors, and potential future work in these areas.

**Response:**

Regarding the accuracy of AIS data, despite the cleaning process described in Section 2.1.2 to improve AIS data quality in terms of time and spatial accuracy,

uncertainties still persist in the following aspects: (1) AIS data gaps and anomalies, which may arise due to methodological conditions, equipment maintenance, etc. (2) Coverage of AIS data. Some vessels, particularly small inland ships and fishing vessels, do not have AIS equipment installed, and their emissions are not included in this study's results, potentially leading to an underestimation of the emissions.

Regarding emission factors, uncertainties stem mainly from two aspects: (1) The uncertainty of emission factor calculations. The emission factors used are fleet-wide averages based on different fuel types, with relatively coarse consideration of fleet composition and operational conditions, which may also be influenced by meteorological and ocean current conditions. (2) Uncertainty in the sources of emission factors. When testing emission factors experimentally, uncertainties can arise due to the experimental methods and sample selection.

To improve the accuracy and reliability of bottom-up ship emission inventories in the future, the following efforts can be made:

1. AIS-based shipping dynamic database establishment and quality control: By incorporating the latest methods and multi-source data, AIS signal errors or malfunctions can be detected more effectively. For example, existing studies have used trajectory mining techniques to detect abnormal vessel trajectories and identify erroneous AIS signals (Sheng and Yin, 2018), and port scheduling data have been integrated to identify ships' arrival and departure statuses (Heikkilä and Jalkanen, 2023). However, these approaches are still limited to certain routes and ports. Future research could expand the application of these methods and data to larger-scale studies.

2. Emission factors: Future work should integrate more recent studies on ship emission factors, particularly the results from tests conducted after 2020. Enhancing the dynamic nature of emission factors and improving coverage for different ship types and sizes is essential.

3. Inventory validation: Future validation could involve integrating multiple data sources, such as satellite observations, air quality monitoring data, and experimental testing data, to refine real-time ship emission algorithms and improve the accuracy of emission inventories.

We have added sentences in the revised manuscript addressing the uncertainties and limitations of our model, as well as potential future work in these areas.

**Revisions in manuscript:**

*Line 545-550*: Although the complex quality control processes employed in this study, uncertainties still persist in the aspects of AIS data accuracy, emission factors and so on. In the next steps, more work should be done to reduce the uncertainties in bottom-up ship emission evaluation model, including integrating latest methods and multi-source data to improve the accuracy of AIS data quality control, gathering more studies on recent ship emission factors to cover more

ship size and operating status, as well as involving multiple data sources such as satellite data to validate the results.

**Comment #3:**

Finally, in the conclusion section, it would be beneficial if the authors could emphasize the global impacts revealed by the spatial heterogeneity of emissions structure and intensity shown in the high-resolution ship emission inventories. For example, how might this spatial variation affect environmental impacts, or what insights could it provide for emission reduction strategies?

**Response:**

The findings on the spatial heterogeneity of global ship emissions offer the following insights for environmental impact studies and policy-making:

1. Tailored management measures: The variation in emission contributions across different vessel types and regions suggests that region-specific emission reduction strategies could be more effective. For example, container ships significantly contribute to emissions in the North Pacific Ocean, South China Sea, and East China Sea. Therefore, focusing on cleaner technologies and operational efficiency for container vessels in these regions would have a substantial impact. The same approach applies to bulk carrier emissions in the Indian Ocean and South Atlantic, as well as Ro-Ro vessel emissions near Europe (e.g., Baltic Sea, North Sea, Mediterranean), etc.

2. Attention to emission hotspots: The Yellow Sea, Persian Gulf, East China Sea, North Sea, and Tyrrhenian Sea are not only areas with the highest emission intensity but also coastal regions with dense populations and ecosystems vulnerable to pollution. This suggests that these regions should be prioritized in environmental management efforts for improving air quality, protecting marine ecosystems, and climate mitigation.

3. Promoting international cooperation to reduce ship emissions: Many high-emission regions include transboundary areas, such as the South China Sea and the Mediterranean, where maritime traffic connects multiple countries. Thus, effective mitigation in these regions will require international cooperation.

We have incorporated the above insights into the revised manuscript.

**Revisions in manuscript:**

*Line 527-531*: The regions with the highest $CO_2$ and $NO_x$ ship emission intensities were the Yellow Sea, the Persian Gulf, the East China Sea, the North Sea, and the Tyrrhenian Sea. These are not only areas with the highest emission intensity but also coastal regions with dense populations and ecosystems

vulnerable to pollution. This suggests that these regions should be prioritized in environmental management efforts for improving air quality, protecting marine ecosystems, and climate mitigation.

*Line 540-544*: The findings on the spatial heterogeneity of global ship emissions offer insights into region-specific management. In addition, since many high-emission regions include transboundary areas, such as the South China Sea and the Mediterranean, where maritime traffic connects multiple countries., effective mitigation in these regions will require international cooperation.

**References**

1. Chen, D., Wang, X., Li, Y., Lang, J., Zhou, Y., Guo, X., and Zhao, Y.: High-spatiotemporal-resolution ship emission inventory of China based on AIS data in 2014, Science of The Total Environment, 609, 776-787, 10.1016/j.scitotenv.2017.07.051, 2017.

2. Crippa, M., Guizzardi, D., Solazzo, E., Muntean, M., Schaaf, E., Monforti-Ferrario: GHG emissions of all world countries - 2021 Report, Luxembourg, 2021.

3. Heikkilä, M. and Jalkanen, J.-P.: The Association between Vessel Departures and Air Pollution in Helsinki Port Area 2016–2021, 10.3390/atmos14040757, 2023.

4. IMO: Emission Control Areas (ECAs) designated under MARPOL Annex VI, 2023.

5. Jasper Faber, S. H., Shuang Zhang, Paula Pereda, Bryan Comer: Forth IMO Greenhouse gas study, London, 2020.

6. McDuffie, E. E., Smith, S. J., O'Rourke, P., Tibrewal, K., Venkataraman, C., Marais, E. A., Zheng, B., Crippa, M., Brauer, M., and Martin, R. V.: A global anthropogenic emission inventory of atmospheric pollutants from sector- and fuel-specific sources (1970–2017): an application of the Community Emissions Data System (CEDS), Earth System Science Data, 12, 3413-3442, 10.5194/essd-12-3413-2020, 2020.

7. Sheng, P. and Yin, J.: Extracting Shipping Route Patterns by Trajectory Clustering Model Based on Automatic Identification System Data, 10.3390/su10072327, 2018.

8. Wang, X., Yi, W., Lv, Z., Deng, F., Zheng, S., Xu, H., Zhao, J., Liu, H., and He, K.: Ship emissions around China under gradually promoted control policies from 2016 to 2019, Atmospheric Chemistry and Physics, 21, 13835-13853, 10.5194/acp-21-13835-2021, 2021.